



# Surging glaciers in High Mountain Asia between 1986 and 2021

Meiping Sun[1,2], Sugang Zhou[1], Xiaojun Yao[1], Hongyu Duan[1], Yuan Zhang[1]

[1]College of Geography and Environmental Science, Northwest Normal University, Lanzhou, 730070, China
[2]Northwest Institute of Eco-Environment and Resources, Chinese Academy of Sciences, Lanzhou, 730000, China

*Correspondence to*: Xiaojun Yao (xj_yao@nwnu.edu.cn)

**Abstract.** High Mountain Asia (HMA) is one of the main distribution areas of surging glaciers in the world. The glacier change represented by the Karakoram anomaly has been a topic of wide concern. Surging glaciers not only reshape the peri-glacial landscape, but may also directly or indirectly lead to disasters. Therefore, it is of great importance to understand the distribution characteristics, periodic laws, and occurrence mechanisms of surging glaciers. Based on Landsat TM/ETM+/OLI

remote sensing images from 1986-2021, a total of 244 surging glaciers in HMA were identified, covering an area of 10974.25 km$^2$, accounting for 11.25% of the total area of glaciers in HMA. The Karakoram Range and Pamirs are the main mountain/plateau where 185 surging glaciers are more highly concentrated. Three basins, including Amu Darya, Tarim and Indus, have 208 surging glaciers comprising 85.25% of the total amount of surging glaciers in HMA, covering an area of 10096.89 km$^2$. From 1986-2021, these surging glaciers advanced at least 2802 times, and exhibited different temporal and

spatial trends. Obvious differences exist in the surge phase and the quiescent phase of glaciers in different regions of HMA. The surge phase of surging glaciers in the Karakoram Range and Pamirs is generally short, mostly between 2~6a, the quiescent phase is 5~19a, and the surging period is 9~24a. The mechanism of surging glaciers in HMA is more complex, which is different from Svalbard and Alaska glaciers.

## 1 Introduction

As an important part of the cryosphere, mountain glaciers are solid reservoirs of precious freshwater resources (Shi and Liu, 2000) and a sensitive indicator of climate change (Oerlemans et al., 1998; Shi and Liu, 2000; Kargel et al., 2014). According to the *Special Report on Global Warming of 1.5℃* released by the IPCC, the temperature in western North America, the European Alps, and High Mountain Asia (HMA) has increased at a rate of 0.3 ± 0.2℃/10a in recent decades, which is higher than the global average warming rate of 0.2 ± 0.1℃/10a (Kang et al., 2020). The intensification of warming made the

glaciers shrink rapidly in HMA (Yao et al., 2019a), with a mass loss of 16.3 ±3.5 Gt/a (0.18 ± 0.04 m w.e.yr$^{-1}$) (Brun et al., 2017). However, the glaciers in some regions of HMA showed different change trends. For example, glaciers in the Nyainqentanglha Mountains shrank rapidly, with a mass change of -0.62 ± 0.23 m w.e.yr$^{-1}$; whereas, the Karakoram Range (-0.03 ± 0.07 m w.e.yr$^{-1}$), Pamirs (-0.08 ± 0.07 m w.e.yr$^{-1}$), Pamir Alai region (-0.04 ± 0.07 m w.e.yr$^{-1}$), and Kunlun Mountains (+0.14 ± 0.08 m w.e.yr$^{-1}$) witnessed relative stability and even advance of glaciers, which is termed the



"Karakoram Anomaly" and has received substantial attention from researchers (Hewitt et al., 2005; Dehecq et al., 2019; Farinotti et al., 2020; Kang et al., 2020).

Glacier surges, also known as exceptional or catastrophic advances, galloping glaciers or pulsatory glaciers, mean that the glacier periodically moves rapidly in a short period of time (Meier and Post, 1969). Surge-type glaciers oscillate between two stages of motion: the quiescent phase and the surge phase (Eisen et al., 2001). The surge phase, characterized by
comparatively fast motion, occurs at quasiperiodic multi-year intervals (Raymond, 1987). In the surge phase, the surging glacier suddenly accelerates and results in a rapid transfer of ice mass, which causes ice displacement while the total mass remains unchanged, and possesses obvious characteristics, such as chaotically crevassed surfaces, rapidly opening crevasses, sheared margins and bulging, overriding, and advancing fronts (Meier and Post, 1969; Zhang et al., 2016). More importantly, the surging glacier poses a great threat to the downstream infrastructure and residents' lives and property. Its rapid
movement can destroy pastures, roads, bridges, villages, hydropower stations, and other facilities in its path in a short period of time (Richardson and Reynolds, 2000; Ding et al., 2018), and even induce glacial lake outburst floods or block rivers to form dammed lakes (Hewitt and Liu, 2010; Yao et al., 2014; Rashid et al., 2020).

Although the number of surging glaciers accounts for only 1% of the total global glaciers, glacier surges have been observed in many parts of the world, including: Svalbard and East Greenland, Norway (Jiskoot et al., 2003); Yukon Territory,
Canada; Alaska, U.S.A. (Copland et al., 2003); and HMA (Sevestre and Benn, 2015; Vale et al., 2021; Guillet et al., 2022). The HMA comprises the largest glacier concentration after the polar regions (Xie et al., 2009; Liu et al., 2017), and is also the most developed area of the cryosphere in the middle and low latitudes. In recent years, surging glaciers in HMA have attracted extensive attention, and some surging glaciers in the Karakoram Range (Gardner and Hewitt, 1990; Barrand and Murray, 2006; Quincey et al., 2011; Bolch et al., 2017), the Pamirs (Osipova et al., 1998; Kotlyakov et al., 2008; Shangguan
et al., 2016; Lv et al., 2019; Goerlich et al., 2020), West Kunlun (Yasuda and Furuya, 2016; Chudley and Willis, 2018; Muhammad and Tian, 2020), Tien Shan (Mukherjee et al., 2017; Zhou et al., 2021), and the Tanggula Mountains (Gao et al., 2021; King et al., 2021; Xu et al., 2021) have been identified. Due to the complex terrain and climate conditions in this region and the unpredictability of glacier surges, most extant literature is based on satellite remote sensing data. The first surging glaciers inventory of HMA (1861-2013) was completed by Sevestre and Benn (2015), and is mainly comprised of
data from former studies. Recently, Vale et al. (2021) identified 137 glaciers as surging between 1987-2019 in HMA by using the GEE platform and the GEEDiT tool. Guillet et al. (2022) presented a regionally-resolved inventory of surge-type glaciers based on their behavior across HMA between 2000-2018, and identified 666 surging glaciers using a multi-factor remote sensing approach that combined yearly ITS_LIVE velocity data, DEM differences, and very-high-resolution imagery (Bing Maps, Google Earth). Due to the differences in methods and periods in the above researches, the number of surging
glaciers identified is not consistent, which makes recognition of the distribution laws and occurrence mechanisms of surging glaciers in HMA insufficient. The main objectives of this paper are three-fold: (1) to identify surging glaciers in HMA between 1986-2021 based on long time series Landsat remote sensing images; (2) to determine the occurrence time and



frequency of these surging glaciers based on available yearly Landsat remote sensing images; and (3) to explore the periodicity and mechanisms of surging glaciers in HMA.

**2 Study area**

High Mountain Asia (HMA) (25°N~46°N, 67°E~103°E) mainly comprises the Qinghai-Tibet Plateau and surrounding alpine ranges, including the Himalayas, Hindu Kush, Karakoram Range, Pamirs, Tien Shan, Kunlun Mountains, Altun Shan, Qilian Shan, Nyainqentanglha Mountains, and Hengduan Shan (Fig. 1), with an average altitude above 4000 m (Xie et al., 2009; Li et al., 2021b). The HMA covers the Xinjiang Uygur Autonomous Region, the Tibet Autonomous Region, Qinghai Province,

Gansu Province, Sichuan Province and Yunnan Province in China, all of Nepal and Bhutan, and parts of Pakistan, India, Afghanistan, Tajikistan, and Kyrgyzstan.

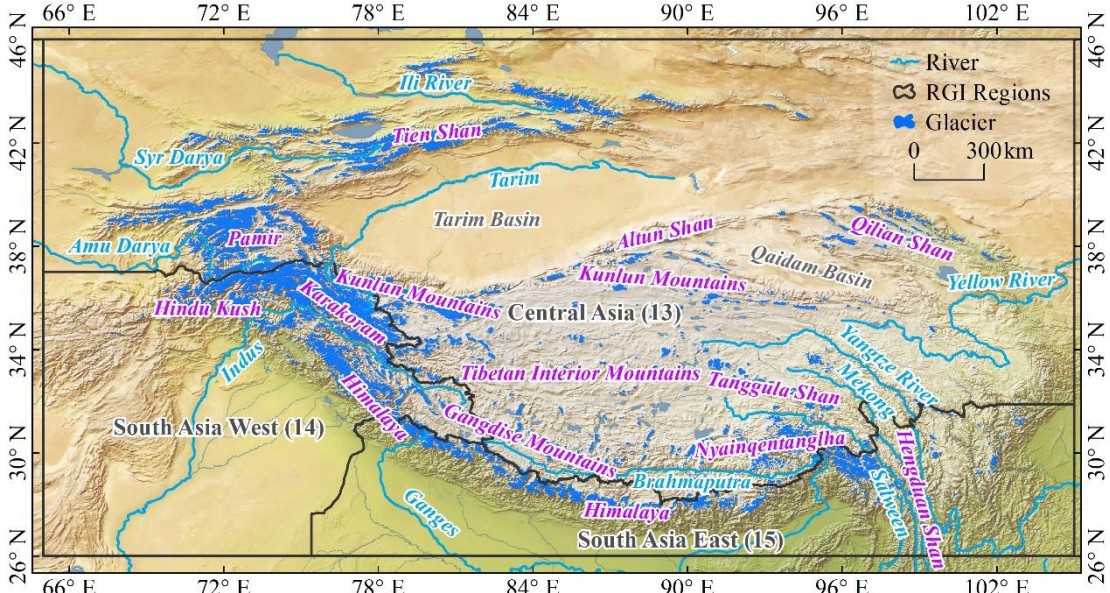

**Figure 1.** High Mountain Asia (HMA).

As an enormous geographical unit, HMA possesses special geographical location and terrain conditions. The southern

part is influenced by Indian monsoons, the eastern and southeastern parts are controlled by East Asian monsoons, and the western and northern parts are influenced by westerly circulation. In addition, plateau monsoons and local small circulation formed by the uplift of the Qinghai-Tibet Plateau play a role in the local climate (Xie et al., 2009). The HMA functions as a water distribution system, termed the Asian Water Tower (AWT), that delivers water to almost two billion people. Abundant glacier ice reservoirs and alpine lakes supply freshwater, and an extended river system encompassing the Yellow, Yangtze,

Indus, Mekong, Salween, Ganges, Brahmaputra, Amu Darya, Syr Darya, and Tarim rivers supply freshwater to downstream areas (Liu et al., 2017; Yao et al., 2022). According to the regional division scheme of the Randolph Glacier Inventory (RGI) V6.0, the HMA involves three regions: Central Asia (13), South Asia West (14) and South Asia East (15), with a total of



95,536 glaciers, covering an area of 97,605.82 km$^2$, accounting for 44.13% and 13.08% of the corresponding amount of global glaciers, respectively (Fig. 1). Based on glacier temperature, surface velocity and climatic conditions, the glaciers in HMA can be divided into three types, including marine glaciers, subcontinental glaciers, and continental glaciers (Shi and Liu, 2000).

## 3 Data and methods

### 3.1 Data sources

#### 3.1.1 Glacier inventory

The Randolph Glacier Inventory (RGI) is a globally complete inventory of glacier outlines. It is supplemental to the database compiled by the Global Land Ice Measurements from Space initiative (GLIMS). Production of the RGI was motivated by the preparation of the Fifth Assessment Report of the Intergovernmental Panel on Climate Change (IPCC AR5). The first version of the RGI was completed and published in 2012. Up to now, six versions of the RGI have been published (RGI consortium, 2017). The RGI data used in this study are version 6.0, which can be downloaded from the GLIMS website (https://www.glims.org/). Most glaciers in China in RGI V6.0 adopt the Second Chinese Glacier Inventory (SCGI), which is the most authoritative glacier dataset in China (Guo et al., 2015; Liu et al., 2015). In the Karakoram region, unpublished glacier data completed by the Technical University of Dresden and the University of Zurich were employed in RGI V6.0. Unfortunately, the vectorized boundaries of glaciers in the Karakoram Range were extracted by a machine classification method and were not manually revised, which caused some snow patches to be included. Therefore, we adopted the SCGI to replace some glaciers that intersected with those in the former dataset. The GAMDAM glacier inventory, completed by Nuimura et al. (2015), covers all glaciers in HMA. However, they exclude thin ice on headwalls and tend to have areas smaller than those measured in conformance with GLIMS guidelines (Raup and Khalsa, 2010). In this study, the GAMDAM dataset is used as a reference for glaciers in southeastern Tibet and other regions in HMA.

#### 3.1.2 Landsat series satellite remote sensing images

Remote sensing images from Landsat satellite series are the most abundant Earth observation data. The remote sensing images used in this study are Level-1 Landsat TM/ETM+/OLI data from 1986-2021, downloaded from the USGS website (https://earthexplorer.usgs.gov/). All available Landsat TM/ETM+/OLI images have been collected in each year and month since 1986. Affected by cloud or snow cover, Landsat images are not available in a few years. After checking one by one, those that can clearly identify the glacier fronts were chosen to extract glacier boundaries. Finally, a total of 7909 Landsat images were obtained, and 112 Landsat satellite orbits were involved (Fig. 2). There are 3419 Landsat TM images, 2417 Landsat ETM+ images, and 2073 Landsat OLI images, respectively. All images were processed with radiometric correction and geometric correction, and topographic correction was performed based on DEM data. For Landsat ETM+/OLI images,





the image fusion of visible band and panchromatic band was carried out, and the spatial resolution of pan-sharpened images was 15 m.

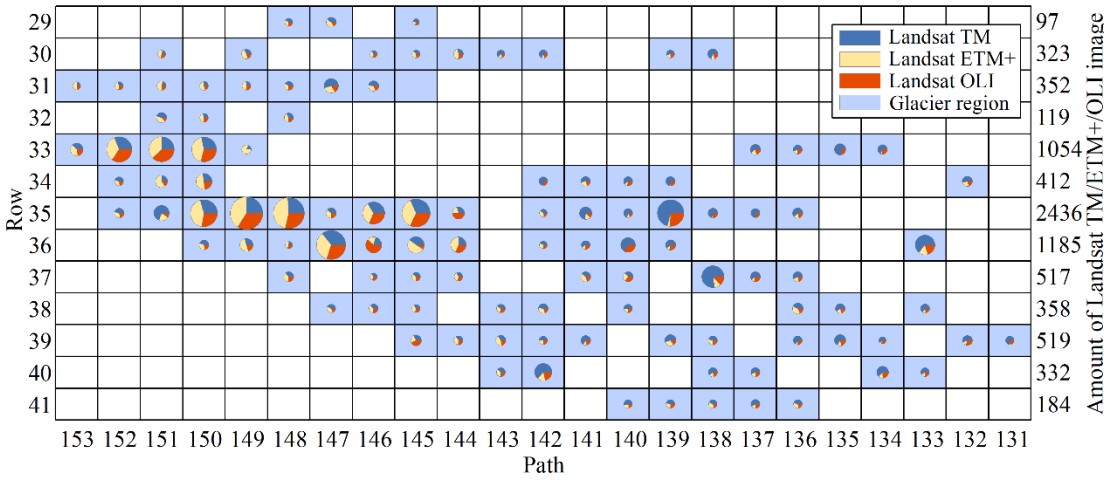

**Figure 2.** Landsat TM/ETM+/OLI images used to monitor surging glaciers in HMA.

### 3.1.3 DEM data

In 2000, the Space Shuttle Endeavour carried synthetic aperture radar and interferometry to capture topography at 1 arc-second (30 m) for over 80% of the Earth's surface. Prior to September 2014, the best available Space Shuttle Radar Topography Mission (SRTM) digital elevation model (DEM) was a 90-m resolution. After several revisions, a SRTM V3 version was produced with a spatial resolution of 30 m and a vertical accuracy of 16 m, which was used in this study. AST14DEM products are DEM datasets based on ASTER stereo images with a spatial resolution of 30 m, which were used to calculate the changes of glacier surface elevation before and after the glacier surge, in order to prove whether or not the glacier surge occurred. Both kinds of DEM data were downloaded from the NASA EARTH DATA website (https://earthdata.nasa.gov/). In the identification process of some surging glaciers, we obtained AST14DEM data of multiple periods.

### 3.1.4 Glacial velocity data

The ITS_LIVE (Inter-Mission Time Series of Land Ice Velocity and Elevation) data product is a part of the NASA 2017 MEaSUREs (Making Earth System Data Records for Use in Research Environments), which provides low-delay measurements of glacier and ice sheet surface velocity and elevation changes from 1985-2018 across the globe. These data are based on Landsat TM/ETM+/OLI images and processed using the auto-Rift algorithm (Gardner et al., 2018), with a spatial resolution of 240 m, downloaded from NASA's Jet Propulsion Laboratory, California Institute of Technology website (https://its-live.jpl.nasa.gov/).





## 3.2 Methods

### 3.2.1 Glacier outline extraction

Numerous methods exist to outline glaciers based on satellite remote sensing images, such as visual interpretation, the ratio threshold method, supervised/unsupervised classification, normalized difference snow index, and the ratio threshold method combined with visual interpretation (Paul et al., 2002; Yao et al., 2012; Paul et al., 2013; He and Yang, 2014; Wu et al., 2017). Among these, the band ratio threshold method based on multi-spectral remote sensing images combined with manual

visual interpretation achieves the highest accuracy (Guo et al., 2015). In this study, the glacier terminal positions were compared one by one in ArcGIS 10.4 to mark the glaciers with significant advance at the end based on Landsat TM/ETM+/OLI images from 1986-2021. Moreover, glacier outlines were revised visually with the aid of RGI V6.0, the SCGI, and GAMDAM datasets. For Landsat TM/ETM+/OLI images, a band composite process was performed using a python code.

The accuracy of glacier outline extraction is mainly affected by remote sensing sensors, image registration (Huggel et al., 2003), and pixel offset errors caused by visual judgment in artificial visual interpretation (Yao et al., 2012; Paul et al., 2013; Sun et al., 2018). Here, we just consider the error resultant from spatial resolution of remote sensing images, which can be calculated by the following equation:

$$\varepsilon = N \times A \tag{1}$$

where $\varepsilon$ is area error (m$^2$); $N$ is the perimeter of a glacier outline (m); and $A$ is the side length of a single pixel (30 m, 15 m, and 15 m for Landsat TM/ETM+/OLI images, respectively).

### 3.2.2 Glacier length extraction

As an important attribute of glacier geometry, glacier length constitutes the basis for calculating the retreat and advance distance of a glacier terminal, which can well characterize the change of glaciers and is a key parameter for global glacier

inventory. Based on the automatic extraction method of glaciers centerline proposed by Zhang et al. (2021), we extracted the length data of surging glaciers in HMA, and took the maximum length as the statistical standard (Yao et al., 2015) to calculate the advance distance of surging glaciers.

The extraction accuracy of glacier length depends on the accuracy of the extracted glacier outline and the quality of DEM data. Generally, the influence of the latter on glacier length is negligible (Yao et al., 2015; Zhang et al., 2021).

Therefore, the spatial resolution of a remote sensing image is the only error source that can be calculated by the following equation:

$$\varepsilon = \left(1 - \frac{\lambda}{L}\right) \times 100\% \tag{2}$$



where $\varepsilon$ is the extraction accuracy of glacier length (%); $L$ is the glacier length (m); and $\lambda$ is the spatial resolution of the remote sensing image (30 m, 15 m, and 15 m for Landsat TM/ETM+/OLI images, respectively).

### 3.2.3 Glacier surface elevation change calculation

The elevation difference between two or more DEM data can be considered as the elevation change on the glacier surface, which is affected by the accuracy of DEM data and co-registration error. In this study, the elevation change error was calculated using the method proposed by Bolch et al. (2011):

$$e = \sqrt{SE^2 + MED^2} \tag{3}$$

$$SE = \frac{STDV_{no-glac}}{\sqrt{n}} \tag{4}$$

where $e$ is the error of elevation change; $SE$ is the standard error of the non-glacial area; $MED$ is the average elevation difference of the non-glacial area; $STDV_{no\ glac}$ is the standard deviation of the non-glacial area; and $n$ is the number of pixels contained in the non-glacial area.

### 3.2.4 Surging glaciers identification

In one region, retreating, advancing and surging glaciers are often mixed with each other, which increases the difficulty and inaccuracy of identifying the target glaciers (Lv et al., 2020). Rapid advance of glacier terminus is the most significant feature of a surging glacier in remote sensing identification (Zhang et al., 2018). Meier and Post (1969) asserted that glaciers with a maximum advance distance of more than 150 m/a can be regarded as possible surging glaciers. In addition, surging glaciers are also accompanied by the development of surface crevasses, ice folds, and moraine folds in the middle of the glacier (Meier and Post, 1969; Barrand and Murray, 2006; Xie et al., 2010). Eventually, a large amount of ice mass is transferred after a glacier has surged, and the surface elevation or terminal shape of the glacier will also change (Lv et al., 2020).

In this study, surging glaciers were identified based on the following criteria: (1) one glacier that increases in length by more than 150 m in one year is tentatively identified as a possible surging glacier; (2) the surface characteristics and terminal shape of the possible surging glacier are observed based on remote sensing images, and moraine folds, cracks, and fragmentation on the glacier surface with lobate, teardrop-shaped glacier terminus are used as features to identify surging glaciers; and (3) the surface elevation change of the glacier obtained from DEM data is further used for the identification of surging glaciers, i.e., a significant surface elevation decrease in the accumulation area, but an increase in the receiving area, constitutes clear proof of a surging glacier.





## 4 Results

### 4.1 Number and distribution of surging glaciers in HMA

Based on RGI V6.0, SCGI and GAMDAM datasets, combined with Landsat TM/ETM+/OLI remote sensing images from 1986-2021, we identified a total of 244 surging glaciers in HMA by comparing the terminal positions, glacier surface features, glacier surface elevation changes, glacier flow velocity changes, and terminal shapes. These glaciers have a total area of 10974.25 km$^2$, accounting for 11.25% of the total glacier area in HMA (Fig. 3). It should be noted that the identified surging glaciers are based on individual glaciers from RGI V6.0 data, and not glaciers that merged with downstream glaciers after surging.

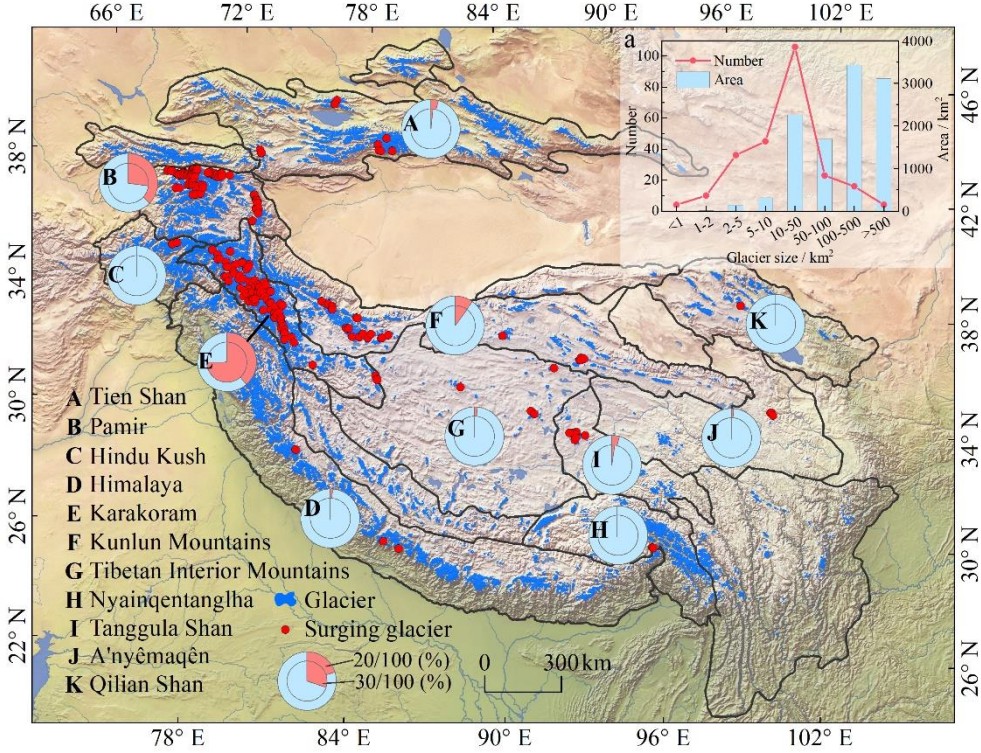

**Figure 3.** Distribution of surging glaciers in HMA.

According to the global mountain system classification scheme (Bolch et al., 2019), the 244 surging glaciers are located in the Karakoram, Pamir, Kunlun, Tien Shan, Tanggula, and Himalayan Mountains (Fig. 3). Among them, a total of 185 surging glaciers with an area of 9445.92 km$^2$ are spatially distributed in clusters in the Karakoram Range and Pamirs, accounting for 75.82% and 86.01% of the total number and area of surging glaciers in HMA, respectively. In the Kunlun Mountains, there are 19 surging glaciers in West Kunlun and five in East Kunlun. There is only one surging glacier in Qilian Shan, and no glaciers in Hengduan Shan and Altun Shan.



In terms of basins, surging glaciers in HMA are mainly distributed in the Amu Darya Basin and Indus River Basin, as well as the inland areas of the Tarim Basin and the Qinghai-Tibet Plateau interior region, and the other six basins have less than 10 surging glaciers (Table 1). Among them, the Amu Darya Basin has the largest number of 79, while the Indus River Basin occupies the largest area of 5439.54 km$^2$. The Amu Darya Basin, Tarim Basin, and the Indus River Basin comprise

208 surging glaciers with an area of 10,096.89 km$^2$, accounting for 85.25% and 91.94% of the total number and area of surging glaciers in HMA, respectively.

**Table 1.** Number and area of surging glaciers in different basins in HMA.

| Basin | Number of glaciers | Number as a percentage (%) | Glacier area (km$^2$) | Area as a percentage (%) |
|---|---|---|---|---|
| Amu Darya | 79 | 32.38 | 2299.30 | 20.94 |
| Tarim | 66 | 27.04 | 2358.05 | 21.47 |
| Indus | 63 | 25.82 | 5439.54 | 49.53 |
| Tibetan Plateau interior region | 14 | 5.74 | 507.89 | 4.62 |
| Yangtze River | 6 | 2.46 | 128.11 | 1.17 |
| Qaidam | 4 | 1.64 | 134.09 | 1.22 |
| Ganges | 4 | 1.64 | 56.19 | 0.51 |
| Yellow River | 3 | 1.23 | 36.50 | 0.33 |
| Syr Darya | 3 | 1.23 | 9.03 | 0.08 |
| Ili River | 2 | 0.82 | 13.73 | 0.13 |

In terms of area size (Fig. 3), surging glaciers with an area of 10-50 km$^2$ (106 glaciers) have the largest number, accounting for 47.32% of the total number of surging glaciers, followed by those with an area of 5-10 km$^2$ (45 glaciers).

Moreover, the total area of surging glaciers with an area >100 km$^2$ (6543.06 km$^2$) accounts for 83.25% of the total area of surging glaciers in HMA, followed by those of 10-100 km$^2$, showing that most of the surging glaciers in HMA are of the medium-to-large scale.

## 4.2 Frequency of glacier surges in HMA

Based on the available remote sensing images, the results show that a total of 2802 advances occurred on 244 surging

glaciers in HMA from 1986-2021 (Fig. 4). The advance of surging glaciers occurred in every year of the study period. Prior to 1993, the annual times of glacier surges in HMA were less than 40 and increased rapidly from 1993-2016, with annual times above 80, except for 40 times in 1995 due to image quality; then, a gradual decrease occurred after 2016. Among them, the surging glaciers advanced most frequently in 2010, 2011, 2004 and 2016, with times of 116, 113, 113, and 112, respectively. This is followed by 2014, 1994, 2006, 2008 and 2007, having times of 109, 108, 107, 106, and 105,

respectively. The fewest occurrences of advances occurred in 1988, at only four times. In terms of months, glacier surges occurred in all months, with the most frequent advances in September and August (526 and 454, respectively), followed by July and October (383 and 335, respectively), and January and February (59 and 71, respectively) being the least frequent. It should be emphasized that the higher frequency of glacier surge found in July, August, September, and October is strongly




correlated to the quality of the images which are better in the vast majority of HMA in these four months, while image
quality is poor in January and February due to heavy snow cover, thus reducing availability.

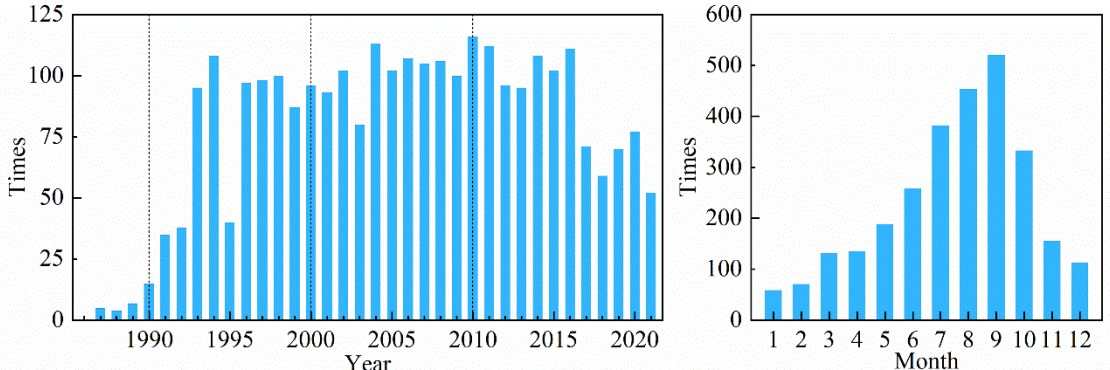

**Figure 4.** Frequency distribution of advances of surging glaciers occurring in different years and months in HMA.

For different mountain/plateau regions, the greatest number of advances of surging glaciers occurred in the Karakoram Mountains (1401), followed by the Pamirs (902) and Kunlun Mountains (261). Fewer advances took place in other regions,
with all below 100 (Fig. 5). Glacier surging events occurred annually in the Pamirs and Karakoram Mountains, with a rapid increase in the Pamirs in 1993 and an annual occurrence above 20 times from 1993-2017, and then this decreased year by year thereafter. Moreover, glacier surging events in the Karakoram Mountains have increased year by year since 1988, and after 1993 (except 1995) the annual numbers all exceeded 30. The Kunlun Mountains experienced fewer surging events until 2007, and then increased year by year, reaching a maximum in 2015, and slowing down afterwards. No glacier surging
events occurred in the Tien Shan and Tanggula Shan after 2010 and 2015, respectively. Overall, the high incidence of surging glaciers in these five mountains is concentrated between 1995-2010, with a decreasing trend of surging glaciers in the last 5-10 years in each region.

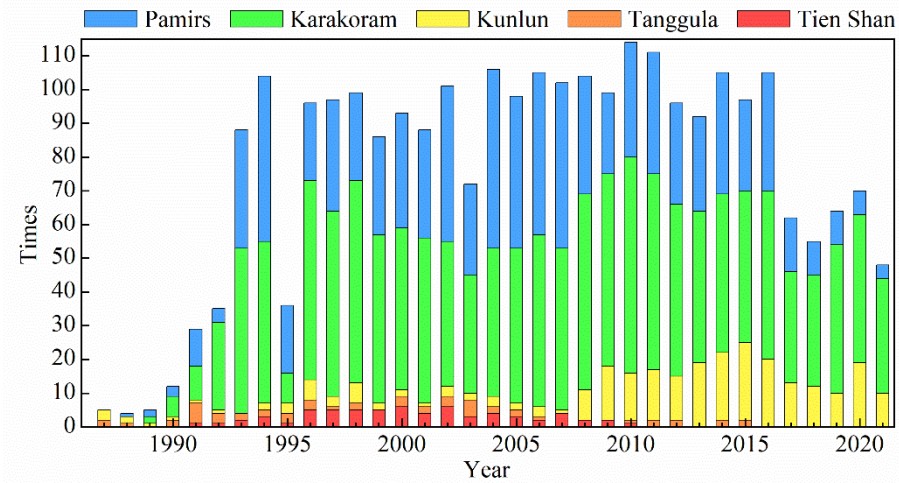

**Figure 5.** Distribution of the number of glacier advances in the main regions of HMA from 1986-2021.





## 4.3 Occurrence characteristics of typical surging glaciers in HMA

To clarify the occurrence characteristics of surging glaciers in HMA, the Oshanina Glacier with the most terminal advance and the Musta Glacier with the most branches in HMA were selected as target glaciers. We analyzed the terminal changes in surge period, as well as the changes in surface elevation and flow velocity of these two glaciers before and after their surges.

### 4.3.1 Oshanina Glacier

The Oshanina Glacier (G071487E39039N) is located in the West Pamir region, having a length and area of 16.58 km and 20.12 km$^2$ in 2000, respectively. Between October 2010 and August 2011, the Oshanina Glacier advanced rapidly by 8285.71 m (30.35 m/d), increasing its area by 4.85 km$^2$ and making it have the longest advancing distance in HMA during the study period (Fig. 6). From October to December 2010, the Oshanina Glacier experienced an advance of 3949.7 m (61.71 m/d) with an area increase of 2.90 km$^2$, reaching the peak of the surge. Then, from June to July 2011, there was an end advance of 4336 m (18 m/d) and an area expansion of 0.85 km$^2$, reaching the sub-peak of the surge (Fig. 7). Landsat TM/ETM+ images with identifiable terminal positions of the Oshanina Glacier were acquired in all months except March 2011, with two to three available images in individual months, which provides an accurate basis for calculating terminal advance velocities during the month. For example, from 1 to 9 December 2010, the end of the Oshanina Glacier advanced 430.7 m with a speed of 53.84 m/d. Similarly, from 5 to 21 July 2011, the end of the glacier again advanced 825.9 m with a similar speed (53.31 m/d).

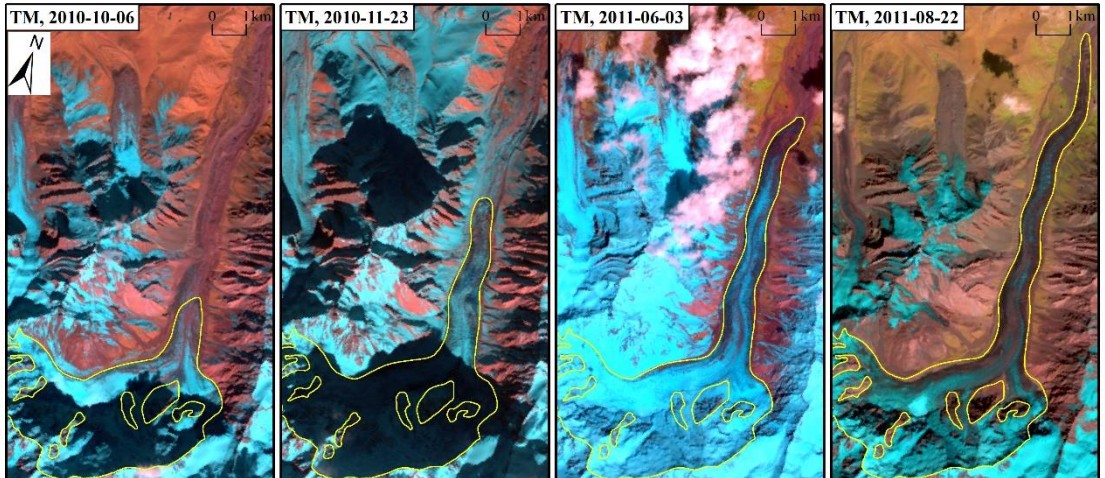

**Figure 6.** Changes of the Oshanina Glacier in the West Pamir from 2010-2011.





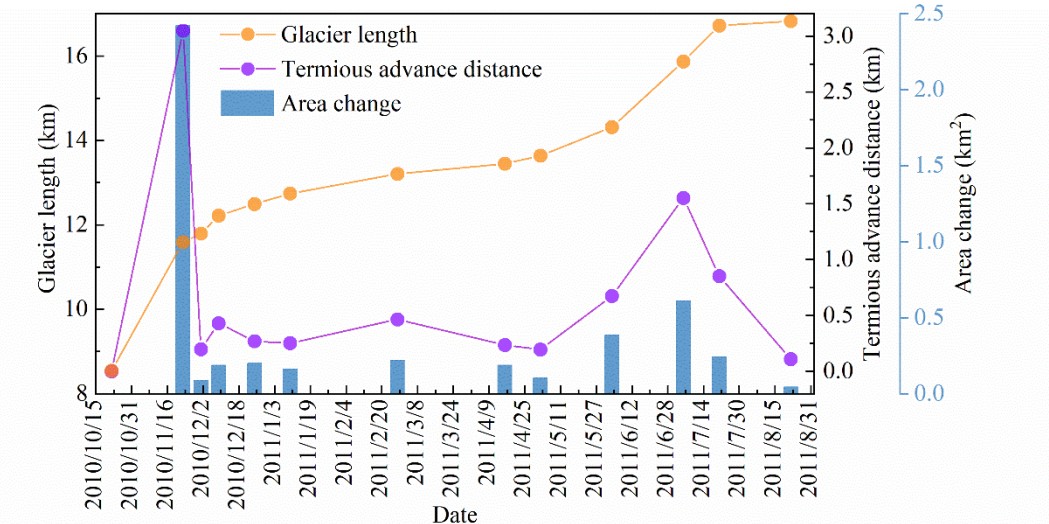

**Figure 7.** Variation in length, terminal advance distance, and area of the Oshanina Glacier in West Pamir during the surging period.

Based on the AST14DEM data obtained in 2009 and 2011, the changes in glacier surface elevation before and after surging of the Oshanina Glacier were calculated, and the results show a clear thinning in the accumulation area and thickening in the receiving area (Fig. 8). The accumulation zone thinned by 67.68 m, on average, and 157 m, at the maximum; whereas, the receiving zone thickened by 80 m, on average, and 168 m, at the maximum. It is evident that the glacier transferred excess ice from the accumulation zone to the receiving zone during the surging process, and the increase

of surface elevation in the receiving zone is consistent with the advance of glacier terminus.

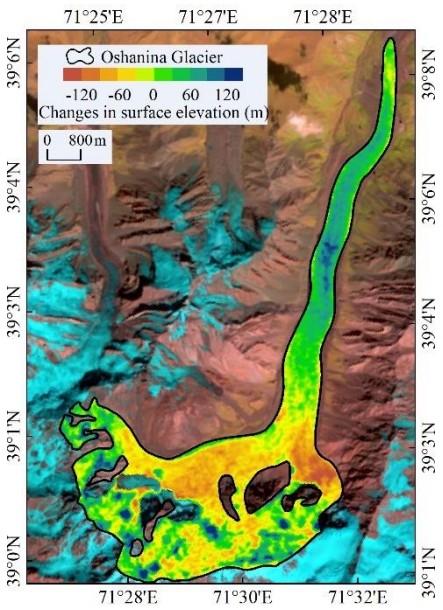

**Figure 8.** Changes in surface elevation of the Oshanina Glacier in the West Pamir from 2009-2011.



### 4.3.2 Musta Glacier

The Musta Glacier, located in the middle reaches of the Kelechin River Basin in the Karakoram Mountains, is a typical
valley glacier with numerous branch glaciers covering an area of 199.16 km$^2$. During the period of 1986-2021, several
branch glaciers (A1, A2, A3, A4) of the Musta Glacier underwent different degrees of surging, but the terminus of the whole
glacier did not change significantly (Fig. 9). Of these, branch glacier A1 started advancing in 1992 and ended in 2000, going
forward a total of 2586.81 m (323.35 m/a) and increasing its area by 1.44 km$^2$. The most significant advance occurred
between 1996-1998, when glacier length and area increased by 1978.95 m (989.48 m/a) and 0.81 km$^2$, respectively, reaching
the surging peak. The end of this branch merges into and overlies the main glacier, with a lobed end and a distinctive push-
back curl (Kotlyakov et al., 1997). Branch glacier A2 started surging the earliest (1991), lasted the longest (21 years), and
advanced slowly from 1991-2004. Then, it advanced rapidly by 557.28 m and increased its area by 0.27 km$^2$ from 2004-2005
prior to entering a deceleration phase. From 1991-2012, this glacier advanced by 2002.27 m, with an increased area of 0.85
km$^2$. Branch glacier A3 moved with the main glacier after merging with it, resulting in a squeezed and curved end.
Compared to the other branch glaciers, branch glacier A3 moved slowly, advancing 1117.2 m with an increased area of 0.32
km$^2$ from 1992-2000. Branch glacier A4 began to surge in 2020, moving forward 1033 m (8.53 m/d), 589.4 m (24.56 m/d),
and 736.9 m (23.03 m/d) from April to August, August to September, and September to October, respectively. Subsequently,
the forward distance and movement speed slowed down to 354.3 m (10.87 m/d) from October to November and 702.4 m
from January to November 2021. From 2020-2021, branch glacier A4 moved forward a total of 3744.2 m and increased its
area by 3.71 km$^2$, and its end converged and overlapped the main glacier, with a broken and fissured surface.



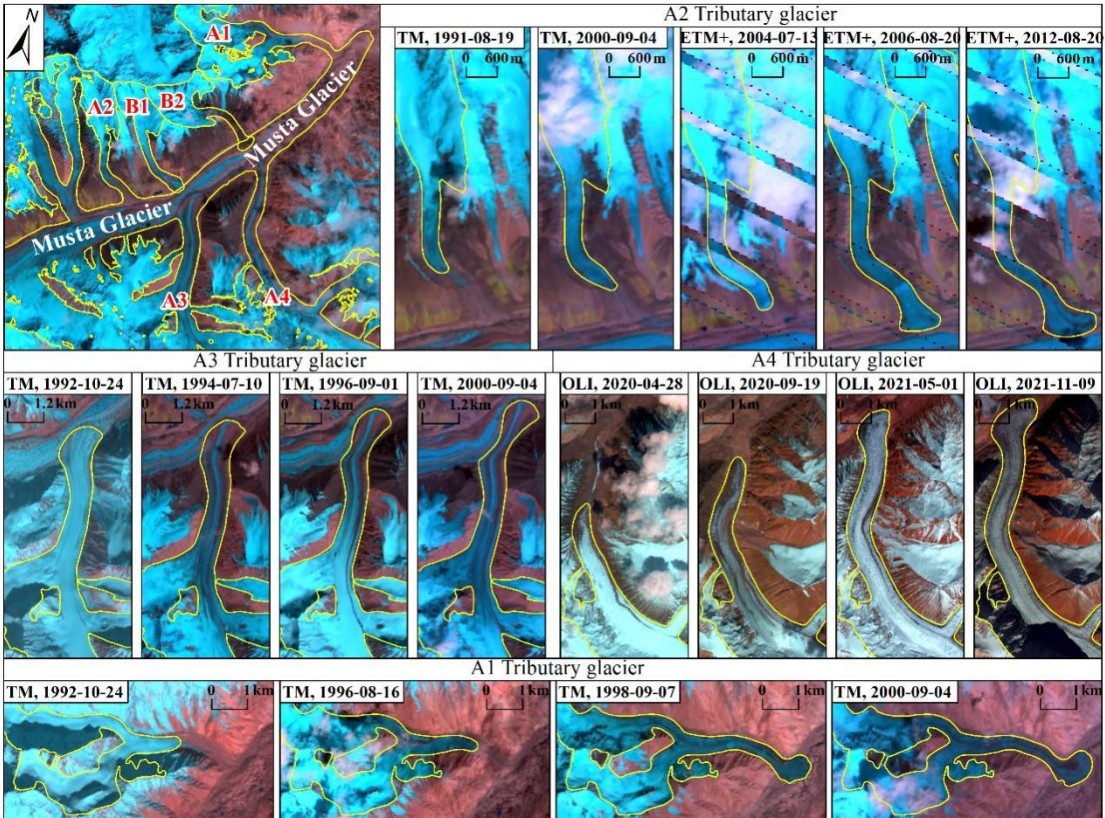

**Figure 9.** Surging process of the Musta Glacier.

Due to the limitation of available data, only changes in surface elevation of branch glaciers A2 and A4 were calculated. The results show that the accumulation and reception zones of the two glaciers had different degrees of thinning and thickening before and after the surge (Fig. 10). The average thinning of the accumulation zone of branch glacier A2 from 2000-2014 was 29.65 m, with a maximum value of 76 m. The average thickening of the receiving area was 19.88 m, with a maximum value of 49 m. Branch glacier A4 experienced an average and maximum thinning of 65.16 m and 214 m, respectively, in the accumulation area from 2019-2021, as well as an average and maximum thickening of 123.53 m and 253 m, respectively. It can be seen that branch glacier A4 discharged excessive ice from the accumulation zone to the receiving zone during the active period.

ITS_LIVE streamflow data from 1986-2018 also show the relatively slow velocities of branch glaciers A1 and A2 of the Musta Glacier (Fig. 11). The velocity of branch glacier A1 reached a maximum of 29.17 m/a in 2003-2005, while the end velocity of branch glacier A2 has increased since 2006. Branch glacier A3 had a faster flow velocity than branch glacier A1 and A2, with a higher flow velocity at 2.4 km from the end in 1989-2000, reaching a maximum velocity of 124.26 m/a. After 2011, there were higher flow velocities at 2.4-6 km from the end, and an increasing trend was observed. The flow velocity of



branch glacier A4 tended to increase year by year between 4.8 and 7.6 km from the end from 1986 and reached the maximum flow velocity (118.89 m/a) in 2018.

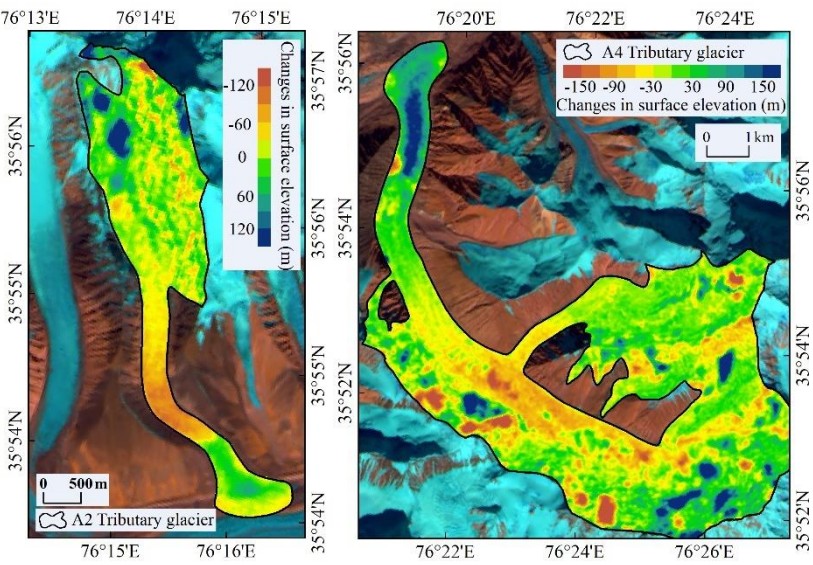

**Figure 10.** Surface elevation change of the Musta Glacier.

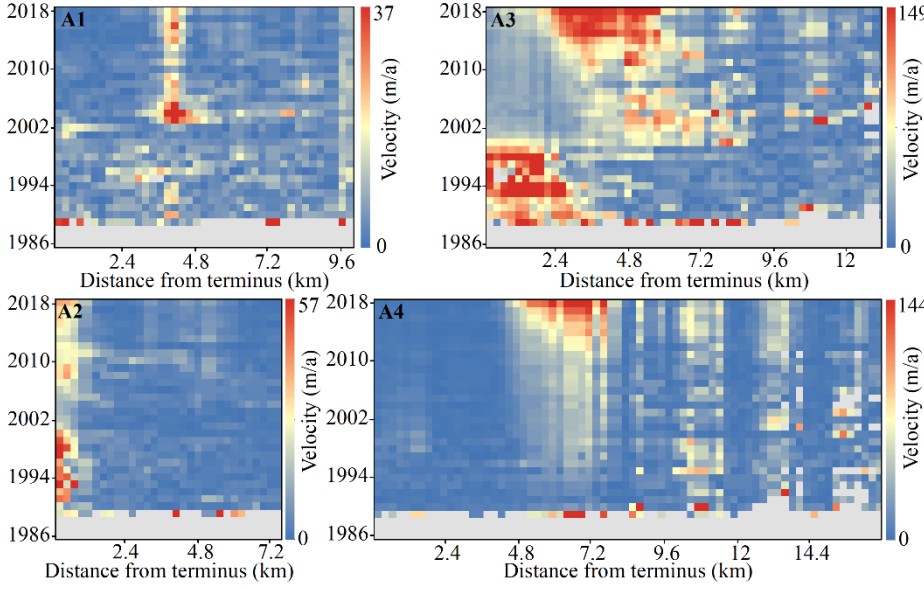


**Figure 11.** Mean annual velocity changes in the centerline of the Musta Glacier.



## 5 Discussion

### 5.1 Similarities and differences with existing surging glacier datasets in HMA

There are three complete (Sevestre and Benn, 2015; Vale et al., 2021; Guillet et al., 2022) and multiple regional inventories
(Kotlyakov et al., 2008; Copland et al., 2011; Rankl et al., 2014; Yasuda and Furuya, 2016; Bhambri et al., 2017; Mukherjee
et al., 2017; Goerlich et al., 2020) of surging glaciers in HMA. However, the number of surging glaciers in HMA varies
widely (Table 2) due to differences in definition criteria, identification methods, study periods, and regions.

**Table 2.** Comparison of the number of surge-type glaciers in HMA identified by other studies with our results.

| Region | Time range | Number | Data source | Identification | Author |
|---|---|---|---|---|---|
| HMA | 1987~2019 | 137 | Landsat TM/OLI | Terminal position of the glacier | Vale et al. (2021) |
| | 2000~2018 | 666 | DEM, ITS_LIVE, Google Earth and Bing Maps | Elevation, velocity, characteristics of the surface glacier | Guillet et al. (2022) |
| | 1861~2013 | 647 | Literature | Multiple methods | RGI V6.0 |
| | 1986~2021 | 244 | Landsat TM/ETM+/OLI, DEM, ITS_LIVE | Elevation, velocity, characteristics of the surface glacier | Our study |
| Pamir | 1861~2013 | 820 | Literature | Multiple methods | Sevestre and Benn (2015) |
| | 1972~2006 | 215 | Resurs-F satellites, Landsat ETM+, ASTER | Elevation, velocity, characteristics of the surface glacier | Kotlyakov et al. (2008) |
| | 1988~2018 | 186 | Landsat MSS/TM/ETM+/OLI, Corona and Hexagon, Google Earth and Bing Maps, SRTM DEM, ASTER GDEM, AW3D30 DEM | The terminal position and surface elevation change of the glacier | Goerlich et al. (2020) |
| | 1986~2021 | 91 | Landsat TM/ETM+/OLI, DEM, ITS_LIVE | Elevation, velocity, characteristics of the surface glacier | Our study |
| Karakorum | 1960~2011 | 90 | Literature, DISP/Keyhole, Landsat MSS/TM/ETM+, JERS-1, ASTER | Surface features, terminal changes | Copland et al. (2011) |
| | 1976~2012 | 101 | Landsat, SAR images | Surface velocity, terminal position | Rankl et al. (2014) |
| | 1861~2013 | 106 | Literature | Multiple methods | Sevestre and Benn (2015) |
| | 1840s~2017 | 221 | Landsat and ASTER | Terminal location, glacier | Bhambri et al. (2017) |




| | | | | velocity, surface characteristics | |
|---|---|---|---|---|---|
| | 1986~2021 | 94 | Landsat TM/ETM+/OLI, DEM, ITS_LIVE | Elevation, velocity, characteristics of the surface glacier | Our study |
| West Kunlun | 1972~2014 | 31 | Landsat MSS/TM/ETM+/OLI, SAR images | Terminal location, glacier velocity, surface characteristics | Yasuda and Furuya (2016) |
| | 1986~2021 | 24 | Landsat TM/ETM+/OLI, DEM, ITS_LIVE | Elevation, velocity, characteristics of the surface glacier | Our study |
| | 1861~2013 | 11 | Literature | Multiple methods | Sevestre and Benn (2015) |
| Tien Shan | 1964~2014 | 39 | Landsat MSS/TM/ETM+/OLI, Corona KH-4, Hexagon KH-9, Cartosat, SPOT | Terminal location, surface characteristics | Mukherjee et al. (2017) |
| | 1986~2021 | 10 | Landsat TM/ETM+/OLI, DEM, ITS_LIVE | Elevation, velocity, characteristics of the surface glacier | Our study |

Sevestre and Benn (2015) were the first to complete a global inventory of surging glaciers, which was recorded in the
RGI data. There are 102 confirmed surging glaciers and 545 possible surging glaciers in HMA in RGI V6.0. In this study,
we identified 57 of the 102 confirmed surging glaciers. Forty-five (45) were not identified because they did not advance
significantly during the study period or advanced a relatively short distance at the end of the glacier. We also identified 46 of
the 545 possible surging glaciers. Four hundred and seventy-five (475) of the 499 unidentified surging glaciers are located in
the Pamir with a relatively small average area of 3.7 km$^2$, in which 82.5% of the total glaciers are less than 5 km$^2$ and 27.2%
are less than 1 km$^2$. Further comparison revealed that most glaciers were unchanged or even retreating at the ends during the
study period, except for some advancing on the surface or end of individual glaciers, which is consistent with Guillet et al.
(2022). In addition, the RGI V6.0 did not include the surging glaciers in the Kunlun Mountains and the Tanggula Mountains,
as well as the Northern Inylchek and Samilowich Glaciers in the Tien Shan that were identified as surging glaciers in some
studies (Mukherjee et al., 2017; Zhou et al., 2021).

Guillet et al. (2022) identified 666 surging glaciers in HMA by comprehensively analyzing the glacier surface elevation
and velocity change, as well as surface characteristics, from 2000-2018, which was significantly higher than our results (244).
A comparison shows that 222 surging glaciers were identified in common. The 443 surging glaciers not identified in this
study had no or only one relatively small glacier end advance. The method used by Guillet et al. (2022) could identify a large
number of glaciers with an advance at the upper part on the glacier surface, but no significant advance at the end, which are
not defined as surging glaciers in this study. Glacier velocity is undoubtedly a reliable basis for identifying surging glaciers;
however, ITS_LIVE data have a low spatial resolution (240 m) and are annual average velocity data, which affects the
identification accuracy of surging glaciers to a certain extent. Furthermore, since Guillet et al. (2022) used the elevation



difference on the glacier surface from 2000-2018, which has a large time span and lacks information in the intermediate years, combined with the large elevation error in high mountain areas, misjudgment may have occurred.

Vale et al. (2021) used GEEDiT, a tool developed by the cloud-based geospatial data platform GEE, to identify 137 surging glaciers from 1987-2019 in HMA, in which 81 surging glaciers were identified by this study. The other 56 glaciers were classified as advancing glaciers, rather than surging glaciers, in this study because of the short advancing distance (less than 150 m) at the glacier terminal. In addition, we identified 161 other surging glaciers with clear evidence of surging.

Scholars have also investigated surging glaciers in some mountains or local regions in HMA. For example, Kotlyakov
et al. (2008) identified 215 surging glaciers in the Pamirs from 1972-2006, while Goerlich et al. (2020) found 206 from the 1960s to 2018, both using the same method as this study. Due to the exclusion of glaciers with small end advance distances by this study, the identified surging glaciers are fewer than those in the above two datasets, with the co-identified surging glaciers of 82. One hundred and four (104) glaciers were not identified as surging glaciers in this study because little to no advancement occurred at their ends, although some of them exhibited a slow advance on the surface. Additionally, we
identified nine other glaciers with significant surging features.

Copland et al. (2011) identified 90 surging glaciers in the Karakoram region from 1960-2011 based on glacier surface features (looped/folded medial, surface foliation, etc.), leading to large errors. Rankl et al. (2014) identified 10 new surging glaciers based on Copland et al.'s result. Bhambri et al. (2017) identified 221 surging glaciers from the 1940s to 2017, but failed to capture some short-term surging glaciers due to the limitations of early available satellite images. In contrast, our
study provides more detailed evidence of surging in the short term.

Yasuda and Furuya (2016) identified nine surging glaciers in West Kunlun, including four glaciers surging from 1972-1992, and five glaciers surging from 1992-2014. A comparison with our result shows that there are six co-identified surging glaciers. Three surging glaciers were not identified in this study because their surging time is beyond the period of this study. In addition, we identified two recent surging glaciers, one of which was identified as a possible surging glacier by Yasuda
and Furuya (2016). Mukherjee et al. (2017) identified 39 surging glaciers in the Tien Shan Mountains based on the literature and available images from 1960-2014; we, however, identified only 10 surging glaciers. The reason for this is that Mukherjee et al. (2017) identified these with slow advance at the end as surging glaciers, while we classified them as advancing glaciers.

Different methods were adopted by the above inventories, and our method results in relatively few surging glaciers.
However, our study provides glacier boundary and length data for each advance of surging glaciers, which constitute direct evidence of glacial surging. Meanwhile, glacier surface elevation and flow velocity were used as supporting evidence for surging glacier identification, which made our results more accurate.

**5.2 Periodicity of surging glaciers in HMA**

A surging cycle is defined as the time between two surges of a glacier, including the surge stage and the recovery stage (Xie
and Liu, 2010). A glacier surge cycle is usually years to several decades, and those of individual glaciers are even longer



(Gao et al., 2019). The surge cycle varies significantly among glaciers in different regions, but the timing of the surge and recovery phases of the same glacier is relatively constant (Björnsson et al., 2003). A total of 36 glaciers in HMA experienced two or more surges from 1986-2021 (Fig. 12), with the most in the Pamirs (19), followed by the Karakoram (13) and Kunlun Mountains (2), and only one each in the Himalayas and A'nyêmaqên Mountains. Based on the surging time of these 36

glaciers and the extant literature, we discussed the surging phase, quiescent phase, and surging cycle of surging glaciers in different regions of HMA.

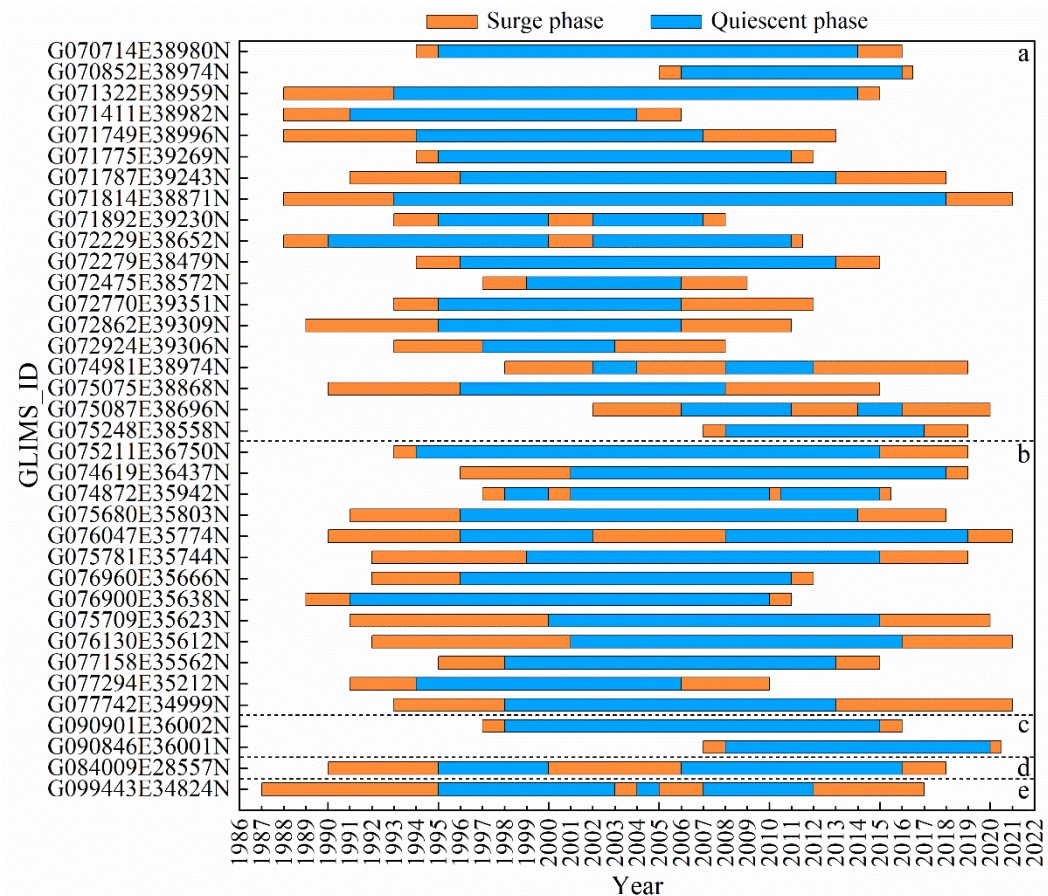

**Figure 12.** The surge phase, quiescent phase, and the time of surge in different regions of HMA from 1986-2021 (orange represents the surge phase, and blue represents the quiescent phase; a-e represents Pamirs, Karakoram, Kunlun Mountains, Himalayas, and A'nyêmaqên

Mountains, respectively).

The surging phase of glaciers in the Pamirs lasts from several months to six years, with longer periods for individual glaciers. Most glaciers in the Pamirs have a surge cycle of 9-20 years and a quiescent period of 5-17 years. For example, the surge period of the Medvezhiy Glacier (G072229E38652N) is approximately 10-14 years, and that of the Bivachny Glacier is approximately 15-20 years (Xie and Liu, 2010; Zhang et al., 2018). The surge phase of glaciers in the Karakoram region is

generally short, ranging from several months to five years. Individual glaciers have long surging periods from 10-24 years,





with quiescent periods ranging from 5-19 years. The surging glaciers in the Kunlun Mountains have surge periods ranging from 1-24 years, and there are only two glaciers with quiescent phases (12 and 17 years) and surge cycles (13 and 18 years). Studies demonstrated that the surging period of glaciers in the West Kunlun region is longer than five years with the surge cycle being longer than 42 years (Yasuda and Furuya, 2016). Guo and Wang (2012) speculated that the surging period of the

Yulinchuan Glacier must be at least more than 40 years. The duration of the surge period in the Tien Shan varies from 1-19 years, with the quiescent phase of 30-50 years or longer and surge cycle of approximately 50 years or more (Zhou et al., 2021). The surging period of glaciers in the Himalayas is two to six years, the quiescent period is five to seven years, and the surge cycle is 10-13 years. The surging period of glaciers in the A'nyêmaqên Mountains is one to eight years, and the Qushi'an No. 17 Glacier (G099443E34824N) has potential ice avalanches, with a quiescent period of eight years and a

surging period of nine years; other glaciers in the region have no previous surge information or records. The surging period of glaciers is two to seven years in the Hindu Kush Mountains, two to eight years in the inner area of the Qinghai-Tibet Plateau, and three years in the Qilian Shan.

**5.3 Climate change of surging glaciers in HMA**

Temperature and precipitation affect melting and accumulation of glaciers, respectively, and their combination determines

the nature, development, and evolution of glaciers (Shi, 2005). Numerous studies have confirmed that the HMA has experienced rapid warming since the 1980s, with a rate unprecedented in the past two millennia (Guo and Wang, 2012; Yao et al., 2019b). According to the observation of meteorological stations, the annual temperature in the HMA from 1979-2020 rose approximately 0.44°C/10a (Zhang et al., 2020; Yan et al., 2020; You et al., 2021), which is much higher than the global warming rate of 0.19°C/10a (Kang et al., 2020). There are also significant seasonal and spatial differences in warming, with

the winter warming rate (0.46°C) being almost twice as fast as the summer warming rate (0.26°C) from 1960-2013 (Yao et al., 2022), and the warming rate increases from south to north and with elevation (Kang et al., 2010; You et al., 2016; You et al., 2021). Yao et al. (2022) obtained an overall increasing trend of precipitation in HMA based on the average of several precipitation data products (3.4 ± 2.8 mm/10a), in which the precipitation in the inflow and outflow regions (Salween, Mekong, Yangtze, and Yellow River Basins) in the northern part of the Qinghai-Tibet Plateau exhibited an increasing trend

(11.0 ± 9.7 mm/10a and 12.3 ± 6.8 mm/10a) and the annual precipitation in the outflow region (Brahmaputra, Ganges, and Indus River Basins) in the southern part decreased significantly (-26.1 ± 11.8 mm/10a). Recently, the mass balance in HMA has generally been negative (Brun et al., 2017; Hugonnet et al., 2021), but is close to equilibrium or positive in local areas, such as the Karakoram, Pamirs, West Kunlun, and Kunlun Mountains. The positive equilibrium means an increased glacial mass accumulation, which could cause changes in the subglacial temperature field and increase the deformation and porosity of the subglacial sediment layer, in turn triggering surging. In contrast, a rapid increase in temperature leading to a negative

of the subglacial sediment layer, in turn triggering surging. In contrast, a rapid increase in temperature leading to a negative mass balance can lead to an increasing glacier melt, which can enter the ice bed through the water system in ice and reduce the friction at the bottom of a glacier, leading to sliding at the bottom of a glacier. In general, the surging glaciers in HMA





have a certain correlation with climate change, but it is difficult to elucidate their surging mechanism based on climate change alone.

**5.4 Possible mechanism of glacier surging in HMA**

Currently, thermal and hydrological mechanisms are the main explanations for the control mechanisms of surging glaciers (Kamb et al., 1985; Murray et al., 2000; Fowler et al., 2017). The thermal mechanism holds that glacier surge is caused by the temperature field change induced subglacial deformation and the increased porosity of the subglacial sediment layer. Based on this, glacier activity is not limited by time and can start and end in any season, and the acceleration and deceleration phases before and after reaching the peak of the surge can last for several years (Clarke et al., 1977). These glaciers usually develop in the Svalbard regions of Norway and Yukon region of Canada, and are therefore called Svalbard-type surging glaciers. The hydrological mechanism asserts that the change from a centralized to a decentralized drainage system at the base of the glacier is the main driver that triggers glacier surges. Surging glaciers under the control of the hydrological mechanism start and end rapidly, usually within days to months. Moreover, these glaciers often start surging in winter when glacier meltwater is less and the subglacial hydrological system is unevenly distributed and poorly drained, and stop in summer when a large amount of meltwater can be used to rebuild effective drainage channels (Björnsson, 1998; Harrison and Post, 2003). They are known as an Alaska-type surging glacier because they usually develop in the Alaska region. The ultimate cause of both of these mechanisms is the enhancement of glacier bottom slip due to increased subglacial hydrostatic pressure.

The surging mechanism of glaciers in different regions of HMA has been widely discussed by researchers. Zhang et al. (2018) reported that the glacier with a GLIMS code of 5Y663L0023 in the East Pamir is more likely to be affected by the thermodynamic mechanism, but the increase in liquid precipitation and ice/snow meltwater will change the hydrological conditions in the ice. Shangguan et al. (2016) and Zhang et al. (2016) suggested that the Karayaylak Glacier in the East Pamir is affected by the thermodynamic mechanism. Li et al. (2021a) stated that the North Kyzkurgan Glacier in the central Pamir is controlled by the thermal mechanism in which the continuous increase of mass in the accumulation zone caused the bottom of the glacier to reach the pressure melting point, triggering the surge. Existing studies present different views on the mechanism of surging glaciers in the Karakoram Range. Quincey et al. (2011) and Hewitt (2007) asserted that the thermal mechanism dominates the surge of Karakoram glaciers, and the increased mass in the accumulation zone at high altitudes results in compression and melting at the glacier bottom. Copland et al. (2011) and Farinotti et al. (2020) stated that changes of hydrological conditions are the main reason for the surge of Karakoram glaciers. In a subsequent observational study, Quincey et al. (2015) reported that the glacier surge in Karakoram is influenced by both hydrological and thermal mechanisms. Lovell et al. (2018) and Iturrizaga (2011) indicated that the high altitude, complex topography, and climatic context combine to produce surging and advancing glaciers in the Karakoram Range. Yasuda and Furuya (2016) held that the combined effect of thermal and hydrological mechanisms causes the glacier to surge in the West Kunlun region. Mackintosh et al. (2017), however, determined that increased glacial meltwater due to climate warming entered the glacier





and led to cracks that triggered glacier surges in the West Kunlun. Häusler et al. (2016) concluded that the surging characteristics of the Northern Inylchek Glacier in the Tien Shan are similar to those in the Karakoram and Alaska. Zhou et al. (2021) indicated that the Samoilwich Glacier may be controlled by the thermal mechanism. Gao et al. (2021) reported that the surge characteristics of glaciers in the Geladandong Peak of Tanggula Shan are not exactly the same as the hydrological and thermal mechanisms, and may be affected by multiple factors.

The above studies only explored the mechanism of partial glacial surges in some regions of HMA, and the results classified surging glaciers as Svalbard or Alaska-type or co-influenced type, but ignored the unique mechanism of surging glaciers in HMA. Surging glaciers of Svalbard and Alaska-types usually occur at high latitudes where the climatic characteristics and topographic conditions differ greatly with HMA located at the high mountains in mid-latitudes, resulting in different responses of glaciers to temperature increase. This study found that there are two main types of surging characteristics for glaciers in the HMA: one is that the surging period lasts for a short time, including two situations: (1) glacier surge peaks at the beginning and completes in a few months; and (2) the surge lasts for several years and occurs in every month, and the surge peak is not limited by season. The second type is that the surge period lasts for a long time, mainly including three situations: (1) the acceleration and deceleration before and after reaching the peak of the pulsation both last for a long time; (2) the surge peaks at the beginning with a long deceleration phase; and (3) there is a long acceleration phase before the start of surging, but the surge stops after the peak and has a short deceleration phase. The above two types are not exactly same as the surging characteristics of Svalbard and Alaska-type glaciers. Therefore, we believe that surging glaciers in HMA have their own unique mechanism, but further proof requires more observational data, and especially field data of interglacial and subglacial structures.

## 6 Conclusions

In this study, we extracted glacier boundary and length information using the manual visual revision method and automatic extraction algorithm of centerline in glaciers based on Landsat TM/ETM+/OLI images. Then, surging glaciers in HMA between 1986-2021 were identified based on typical characteristics of glacier surge. Furthermore, we analyzed the change characteristics of surface elevation and surface velocity of typical surging glaciers combined with DEM and ITS_LIVE glacier velocity and other data to examine the climatic background, surging period, and possible surging mechanism of surging glaciers in HMA. The main conclusions are as follows:

(1) There are 244 surging glaciers in HMA from 1986-2021, covering an area of 10974.25 km$^2$, accounting for 11.25% of the total area of glaciers in HMA. The Karakoram Range and Pamirs are the areas of concentrated surging glaciers (185, 75.8%), which are clustered in space. Two hundred and eight (208) surging glaciers (85.25%) with an area of 10,096.89 km$^2$ (91.94%) are located in the Amu Darya, Tarim, and Indus Basins.

(2) From 1986-2021, at least 2802 glacier advances occurred on 244 surging glaciers in HMA, exhibiting different spatio-temporal trends. The annual surge times occurring in 1993-2016 were all greater than 80, except for the 40 times in





1995 due to image quality, and the surges gradually decreased after 2016. The glaciers in HMA surged in every month, and August and September are the high period of glacier advances, with counts of 526 and 454, respectively. The second-highest

number of glacier advances took place in July and October (383 and 335). January and February (59 and 71, respectively) had the fewest glacier surges.

(3) The surging phase, quiescent phase, and surging cycle of glaciers in HMA vary in different regions. The surging phase of glaciers in the Karakoram Range and Pamir is generally short, mostly between 2~6a, with the quiescent phase of 5~19a and the surging cycle of 9~24a. The mechanism of surging glaciers in HMA is more complex, having characteristics

that are not exactly the same as those of Svalbard and Alaska surging glaciers.

**Data availability.** The dataset of surging glaciers in HMA can be provided upon request from the corresponding readers.

**Author contributions.** MS and SZ wrote the original draft; XY contributed the conceptualization, supervision, funding acquisition, as well as editing of the manuscript; SZ, HD and YZ contributed in terms of getting satellite data and processing data.

**Competing interests.** The authors declare that they have no conflict of interest.

**Financial support.** This work was supported by the National Natural Science Foundation of China (grant nos. 42161027 and 42071089), the Open Research Fund of the National Cryosphere Desert Data Center (grant no. 20D02), the Open Research Fund of the National Earth Observation Data Center (grant no. NODAOP2020007), and Gansu Province Education Science and Technology Innovation Project (grant no. 2021QB-019).

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
