# Peer review of "Surging glaciers in High Mountain Asia between 1986 and 2021"

_EGUsphere, 2022_

## Referee Comment (RC1)

**Review of '*Surging glaciers in High Mountain Asia between 1986 and 2021*' by Sun and co-authors, 2022**

The study by Sun and co-authors presents a new dataset of glacier surges in High Mountain Asia over the period 1986-2021. The surges seem to have been identified manually based mostly on glacier terminus position but also complementary datasets of glacier velocity and elevation change. The manuscript includes some description of surge spatial distribution and cycles.

This is a topical line of research, which tackles a very relevant question for the monitoring of surging glaciers and their underlying processes. However, I have a number of (very) major comments related to the soundness of the methods and results analysis, as well as to the novelty and relevance of the findings, that would need to be addressed for further consideration of this work.

I have also included a number of minor comments that can hopefully help the authors in revising their manuscript. These are however far from complete given the considerable revisions needed.

**Major comments**

**Study area**

This part is very general and none of it feels particularly relevant for the research questions outlined above. Furthermore, there are a number of sentences that need to be backed up with references.

**Data and methods**

There are long descriptions in the data and methods that are irrelevant for the study and/or could be condensed in 1-2 sentences. On the other hand some paragraphs lack crucial details and references. Example:

- 3.1.1. The RGI is commonly used in the scientific community and does not require more than 1 description sentence. On the contrary, the explanations should be focused on the choice of inventory in this study – they currently do a poor job at this and the end of the paragraph is particularly confusing.
- 3.1.2. lacks crucial details and references to be able to evaluate the methods used.
- 3.1.3. and 3.1.4. lack relevance. SRTM data is commonly used and the presentation of this dataset could be summarized in one sentence focusing on how it was used in this study. Similarly for the ITS_LIVE data.
- 3.2.1. belongs to the category of paragraphs that lack clarity. After reading it several times I still do not understand how these glacier terminal positions were identified.
- In 3.2.3. there is a major methodological gap: how were the SRTM and ASTER DEMs co-registered?
- 3.2.4 could (should) be backed up with visual examples.

Furthermore, manually identifying glacier surges across all glaciers in HMA sounds like a huge amount of work that is likely to lead to errors based on the subjectivity of the operator(s). Consistency of the identification across such a large region is also questionable, especially considering the criteria used which are very subjective. More clarity is needed to get a better sense of what actually was done to identify the surges, and above all this mapping effort needs to be validated.

**Results**

This section is long with many parts where the relevance is questionable.

Section 4.2. is strongly influenced by data quality/availability and I am therefore skeptical that one can learn anything from it.

Section 4.3. is interesting but I fail to see how these two glaciers are 'typical' examples of glacier surges in HMA.

There are no uncertainties in the results.

**Discussion**

I find 5.1. hard to follow considering the little information available on the methods used in this study and the lack of validation.

5.2 belongs more to the 'results' section. This part is lacking a discussion part about the implications of these results (and references).

5.3. Does not make any sense. If the authors want to say something about the influence of climate on surges, they need to demonstrate it.

In 5.4, it is worth noting that the timing of the surges alone does not say much about the mechanisms at play. In addition, I am extremely skeptical about the claims made at the end of the paragraph – these would need to be appropriately demonstrated.

**Figures**

Most of the figures are of poor quality and/or hard to read. The captions consist of handful of words that hardly describe the content of the figures.

**Minor comments**

**Title & Abstract**

L6: English could be improved.

L6-7: I do not see the relevance of the Karakorum anomaly here.

L11: Round to only keep significant digits. This is a recurring issue throughout the manuscript.

L12: 'plateau' needs to be removed here and throughout the text.

L12-13: I don't see the relevance of thinking in terms of basins here.

L16: 2-6a -> 2-6 years

L17-18: Could the authors provide more details about this in the abstract?

**Introduction**

L26: An important reference here and for the next sentence would be Hugonnet et al., 2021. Also Miles et al., 2021.

L31: the link here between the Karakorum anomaly and surging glaciers is quite obscure.

L37: 'chaotically crevassed surfaces' and 'rapidly opening crevasses' seems to be a repetition.

L48: Also Bhambri et al., 2017 ; Hewitt et al., 1969 ; 1998

L54: Sevestre and Benn (2015) based their inventory solely on already published datasets. I am not sure this is the best reference here, or the sentence would need to be adapted.

L62: English can be improved

**Study area**

L69-71: I do not see how this is relevant for a study on surging glaciers.

L74: 'enormous' is not really appropriate.

L78: references missing.

**Data and methods**

L102-103: this is contractictory with the previous sentence.

L105: relevance? references?

L106: Acronyms need to be spelled out at their first occurrence.

L108-109: I have a hard time imagining that this was done manually for a collection of tens of thousands of images. What criteria were used to remove the images? Was the entire image removed as soon as there were a few clouds or were the clouds clipped from the images? Why not use some cloud masking algorithms?

L111-112: Additional details on these processing steps are necessary to evaluate the soundness of this approach. Above all, references.

L113: English (plural)

L113: More details needed (parameters used)

L113: Why not simply use the pan-sharpened images? How are the visible bands used? Unclear in what follows.

L121: references missing

L124: references missing

L125-126: You need to specifically mention which periods.

L131: Key missing reference here is the work by Dehecq et al., 2019.

L139-140: Which band ratio are you referring to here? Does this not depend on the sensor resolution?

L140: How were the glacier terminal positions mapped? Manually? With the band ratio approach? How are debris-covered glaciers dealt with?

L143-144: details missing about the algorithm used.

L148: reference missing.

151: It would be appropriate to mention the satellites (Landsat 5, 7, 8) in addition to the sensors.

L162: the '%' sign should be removed from the equation. Also '100' if it's already specified that epsilon is in %.

L160-164: references missing

L166: English could be improved (… between two DEMs…). Why 'two or more' and not simply 'two'?

L175: Which region? This is not clear.

L180: Glasser et al., 2021 could be a relevant reference here.

L184-189: These criteria sound very subjective and it would be good to at least show examples of these changes. The third criteria is dependent on good co-registration of the DEMs. What kind of values were used for the elevation changes?

**Results**

L206-211: I still do not see the point of counting surging glaciers in water basins.

L215-217: Wouldn't it be better to express these in %?

L267: What do you call the 'receiving' area?

L294: What is the 'reception' zone?

**Discussion**

Sevestre and Benn (2015) are based on a number of regional studies. For consistency in the methods, it would make more sense to refer to these regional studies rather than this review paper.

L322: What do you mean by 'significantly'?

L336-339: I am skeptical about this argument (one does see surges in the 2000-2018 dH patterns), especially as the time span used in this study is not specified.

L346: Do you mean manually?

L398: This title does not make any sense

**Figures**

Figure 1:

Source of the glacier outlines? Background image?

Additional content (e.g. location of identified surging glaciers by this study or previous ones) would be welcome to make the figure fit in the study – the relevance of a general map of HMA here is questionable.

Figure 2:

This figure needs a detailed explanation (make use of the caption). I do not find the path number very informative. Rather, it would make more sense to project this table on a map to see the actual image extents.

I am also skeptical that the type of images matters here. It would be more relevant to show the total number of images or the number of images per month for each image extent.

Figure 3:

Some regions are missing a diagram. I do not find the number of glaciers in (a) to be very informative. The caption needs to be filled.

Figure 4:

How is the quality/availability of the data affecting these results?

I am guessing that in (b) this corresponds to the start of the surge? How is this defined?

Figure 7:

It would be nice to extend the x-axis to earlier dates to confirm that no advances occurred earlier.

Table 2:

No need to repeat the global studies in the regional ones. Sevestre and Benn (2015) use the data from other studies so could also be removed from the table.

Figure 12:

Putting the name of the regions directly in the figure would help improve the readability.